# Reinforce Your Layout:
# Online Reward-Guided Diffusion for Layout-to-Image Generation

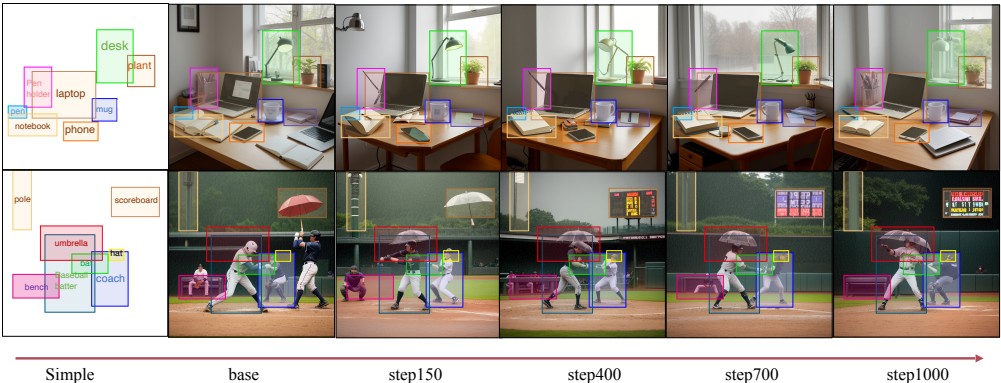

Figure 1: Qualitative visualization of the curriculum-guided reinforcement learning process of ARPO. Each row shows an example where the model starts from a simple layout (left) and progressively refines the generated image as training proceeds (base, step150, step400, step700, step1000). Bounding boxes of different colors denote the target objects. As training advances, object placement becomes increasingly accurate and visually coherent, demonstrating how our method incrementally improves spatial alignment and realism.

## Abstract

In this work, we tackle the layout-to-image generation task by proposing a novel online reinforcement learning (RL) framework that directly optimizes diffusion models to achieve consistency between images and layouts. We introduce RLLay, a method that overcomes a major limitation that lies in existing methods with their reliance on indirect side guidance—rather than direct supervision on layout alignment—which constrains these models' ability to accurately position and scale image content. Given a prompt, our approach generates multiple candidate images and ranks them using a reward model based on Intersection-over-Union (IoU) to quantify alignment between predicted and target layouts. To effectively utilize this ranking signal, we introduce a pairwise preference-based optimization strategy that fine-tunes the diffusion model by maximizing the likelihood of higher-ranked samples relative to lower-ranked ones (hard-negatives). Experimental results show that our RL-based fine-tuning significantly improves both spatial layout fidelity and text-image alignment, establishing a promising direction for more controlled and layout-aware image generation.

## 1 Introduction

With recent advances in text-to-image generative models (Ramesh et al., 2021; Nichol & Dhariwal, 2022; Rombach et al., 2022; Chen et al., 2023; Xue et al., 2024), there has been growing interest in layout-to-image generation (Li et al., 2023a; Wang et al., 2024; Guo et al., 2024) allowing users can explicitly control the spatial locations (Xie et al., 2023; Wang et al., 2024; Li et al., 2023a), and

counts (Binyamin et al., 2024; Yang et al., 2023b) of objects within generated images. While existing approaches achieve satisfactory results, they typically require extensive and accurately annotated bounding-box datasets for supervised training, resulting in significant data collection overhead.

Recent developments in reinforcement learning (RL) and preference-based learning, notably within large language models, have inspired new methods for image generation (Zhu et al., 2025; Xu et al., 2023a). Preference-based learning approaches such as DPO (Fan et al., 2024) offer stable training dynamics with lightweight frameworks compared to traditional RL-based methods (Li et al., 2023b), which often experience instability or suboptimal performance without careful hyperparameter tuning. Nevertheless, current offline preference methods like DPO rely heavily on pre-collected paired data, which is costly to acquire and limits adaptability. Furthermore, the optimization is indirect because preference comparisons often involve externally generated images rather than samples from the model itself, hindering training efficiency and adaptability in dynamic scenarios.

Online preference optimization (Fan et al., 2023; Black et al., 2023) emerges as an appealing alternative by directly optimizing model outputs in real time. However, transitioning preference-based methods to an online setting is nontrivial: generating paired data simultaneously with the training process yields only limited samples at each step, causing gradient updates to become unstable and negatively affecting image quality. This instability arises because gradients calculated from sparse or poorly differentiated image pairs produce noisy and inconsistent updates. We introduce Advanced Relative Policy Optimization (ARPO), which does not rely on any sensitive or learned reward model and yields substantially more stable training in the online setting by directly optimizing pairwise preferences with explicit log-probabilities.

We summarize our contributions below:

1. We develop a **new pipeline called RLLay for layout-to-image generation** using reinforcement learning. To the best of our knowledge, this problem has not been previously addressed.

2. We introduce a novel preference-driven reinforcement learning algorithm **ARPO**, which achieves stable gains even when reward signals are complex and reliable pairwise data are scarce, advancing beyond existing approaches such as DPO, DSPO, and GRPO.

3. In detailed implementations, we propose an enhanced online preference optimization framework featuring  to mitigate training instability under settings with limited online data. We present a hard negative mining strategy, which maximizes the utility of every generated data pair by leveraging the full diversity of online image generation, leading to improved image quality and more robust preference learning.

## 2 RELATED WORK

### 2.1 CONDITIONAL IMAGE GENERATION

Diffusion models have emerged as powerful generative frameworks, excelling in synthesizing high-quality images from textual descriptions. Early foundational work by Ho et al. (Ho et al., 2020) introduced denoising diffusion probabilistic models (DDPM), which inspired subsequent enhancements for text-to-image (T2I) generation such as Stable Diffusion (Rombach et al., 2022), GLIDE (Nichol & Dhariwal, 2022), and Imagen (Saharia et al., 2022). These methods typically employ supervised training objectives with deterministic or stochastic denoising schedules, enabling precise yet flexible image generation conditioned on textual prompts.

Generating images consistent with spatial layouts has been explored extensively, employing various architectures and training strategies. Early CNN-based approaches (Hong et al., 2018; Li et al., 2019) relied on direct regression or adversarial losses to enforce spatial constraints. Transformer-based methods, such as LayoutTransformer (Yang et al., 2021), exploited attention mechanisms to enhance semantic and spatial coherence. Diffusion-based methods have recently emerged, with CreatiLayout (Guo et al., 2024) and Hico (Zhang et al., 2023) effectively leveraging diffusion architectures to improve spatial accuracy and visual realism via bounding box annotations, aiming to produce images that adhere to predefined layouts.. Despite their strengths, these methods largely depend on deterministic training paradigms, limiting their flexibility in explicitly optimizing layout adherence via external reward signals. Our method bridges this gap, combining diffusion models

with RL-driven pairwise preference learning to explicitly and effectively enhance spatial fidelity and textual alignment in layout-to-image generation.

## 2.2 PREFERENCE LEARNING IN GENERATIVE TASKS

Reinforcement Learning (RL) has been effectively utilized in generative modeling to optimize models directly towards user-defined objectives. Early efforts such as GAN-based approaches (SeqGAN (Yu et al., 2017), RankGAN (Lin et al., 2017)) and policy-gradient methods (Ziebart et al., 2008) demonstrated RL's capability to guide generation via explicit reward signals. Recent advances integrate RL with diffusion models to enhance complex conditioning tasks. Black et al. (Black et al., 2023) introduced DDPO, optimizing diffusion models through policy gradients for objectives that are difficult to capture explicitly by standard supervised learning, such as aesthetic quality and alignment with user preferences. Similarly, Fan et al. (Fan et al., 2024) demonstrated significant improvement in text-to-image alignment by incorporating RL-based fine-tuning with KL-divergence regularization.

Pairwise preference learning has become prominent due to its effectiveness in aligning generative models with subjective human judgments. Christiano et al. (Christiano et al., 2017) and Ouyang et al. (Ouyang et al., 2022) initially proposed preference-based methods in reinforcement learning from human feedback (RLHF), popularizing pairwise comparison paradigms. Rafailov et al. (Rafailov et al., 2023) introduced Direct Preference Optimization (DPO), enabling training without explicit reward modeling by directly optimizing preference pairs. Recent extensions such as RankDPO (Karthik et al., 2024) generalized this approach to ranking-based preferences, further enhancing generative models' alignment with nuanced human judgments. In the image generation domain, ImageReward (Xu et al., 2023b) utilized human-rated image pairs to train robust reward models, subsequently guiding diffusion models towards outputs aligned closely with human aesthetic preferences. DSPO (Direct Score Preference Optimization) (Zhu et al., 2025) is a novel fine-tuning algorithm for diffusion-based text-to-image models that aligns the pretraining and preference alignment objectives via score matching, enabling human-preference-consistent image generation without requiring explicit reward models.

## 3 METHOD

This section presents **RLLay** (Reinforce Your Layout): an online reinforcement learning fine-tuning method for diffusion-based layout-to-image models that aims to substantially improve *layout consistency* without sacrificing fidelity or semantic coherence. First (§ 3.1), we formalize the task with a text prompt and layout meta-information as inputs, and define a unified *layout reward* $R_{\text{layout}}$ as the mean IoU between predicted and target boxes in the generated image. Next (§3.2.1), we introduce a curriculum by layout difficulty, training in a progressive easy-to-hard schedule to stabilize optimization and preserve separable preference signals. Then (§3.2.2), to obtain stronger supervision, we generate multiple candidates per (prompt, layout), rank them by $R_{\text{layout}}$, and construct *extreme* preference pairs to amplify reward gaps and log-probability differences, improving learning efficiency under the same inference budget. Additionally (§3.2.3), since policy gradients require *differentiable* action probabilities, we extend SD-3's deterministic Flow–ODE to a variance-time SDE and derive a one-step explicit log-probability $\log \pi_\theta$. Building on this (§3.2.4), we optimize a loss that combines *pairwise logistic advantage* with an *IoU-aware KL* regularizer, yielding a learning signal that is both directional and stable.

## 3.1 PROBLEM FORMULATION

We formulate the task as follows. Given a text prompt $\mathbf{p}$ and a set of layout meta-information

$$\mathcal{M} = \big\{ (b_i, \, t_i) \big\}_{i=1}^{K}, \tag{1}$$

where

$$b_i = \big( x_{1,i}, \, y_{1,i}, \, x_{2,i}, \, y_{2,i} \big)$$

is the bounding box of the $i$-th object specified by its top-left corner $(x_{1,i}, y_{1,i})$ and bottom-right corner $(x_{2,i}, y_{2,i})$. All coordinates are normalized to $[0, 1]$ with the image origin at the top-left $(0, 0)$, the $x$-axis pointing right, and the $y$-axis pointing down. The term $t_i$ denotes the textual description of

the region (e.g., "red car"). Our goal is to generate an image $I$ that is semantically aligned with $\mathbf{p}$ and spatially consistent with $\mathcal{M}$.

To quantify spatial consistency, we compare predicted bounding boxes in the generated image with the target layout. Let $o_i$ be the object instance in $I$ corresponding to description $t_i$, and let $\text{Box}_I(o_i)$ denote its predicted bounding box. We define the layout reward as

$$R_{\text{layout}}(I, \mathcal{M}) = \frac{1}{K} \sum_{i=1}^{K} \text{IoU}\big(\text{Box}_I(o_i), b_i\big). \tag{2}$$

Here, $\text{IoU}(A, B) = \frac{|A \cap B|}{|A \cup B|}$ denotes the Intersection-over-Union between two boxes—a standard measure of spatial overlap in object detection and localization (Everingham et al., 2010).

## 3.2 REINFORCE YOUR LAYOUT

### 3.2.1 CURRICULUM LEARNING BY LAYOUT DIFFICULTY

**Motivation.** Online fine-tuning over the full data distribution is unreliable: at early stages, the base model typically fails to satisfy complex layout constraints, so parallel sampling yields candidate pairs of uniformly poor quality with barely separable preference/reward signals, making policy improvement difficult. In addition, the wide distributional span within a single dataset further amplifies training instability. We therefore adopt a progressive curriculum by layout difficulty: the model first consolidates basic spatial alignment on easy layouts, and is then gradually exposed to harder scenes characterized by higher object counts, stronger overlaps, and tighter inter-object relations. This staged schedule stabilizes optimization while preserving useful preference signal as complexity increases.

**Difficulty factors.** We characterize difficulty using three signals:

- *Box count $K$*: more boxes increase combinatorial and occlusion complexity.
- *Overlap*: measured by average pairwise IoU or total overlap ratio (higher overlap makes localization harder) (Li et al., 2025).
- *Preference bias (co-occurrence overfitting)*: diffusion models tend to overfit frequent co-occurrences in training data (e.g., *baseball player* $\rightarrow$ *bat*), which often overrides rare or counterfactual combinations (e.g., *baseball player with an umbrella*) (Sun et al., 2025; Carlini et al., 2023). This factor captures compositional failures from semantic priors.

**Level definitions.** Based on these factors, we partition samples into four levels:

- **Easy:** $K \in [1, 3]$, Overlap $< 10\%$, no preferences/relations.
- **Medium:** $K \in [3, 4]$, Overlap $< 20\%$, with preferences.
- **Hard:** $K \in [4, 5]$, Overlap $> 30\%$, with preferences and *containment*.
- **Very Hard:** $K > 5$, Overlap $> 30\%$, with preferences, *containment*, and *over four bounding boxes crossing*.

**Scheduling.** We train using stage-wise shuffled mixtures of the four difficulty levels; the exact level definitions and per-stage mixing ratios are reported in Sec. 4.1.2.

**Outcome.** This curriculum first yields stable, separable signals on easy cases, then increases exposure to hard/very-hard layouts so the policy continues to improve under heavy overlap and complex relations without collapsing. The results of the training stage are shown as in Fig 1.

### 3.2.2 HARD NEGATIVE MINING VIA EXTREME PAIRING

For each (prompt, meta) pair, we generate six candidate images in parallel. To assess their spatial alignment quality, we compute the layout reward $R_{\text{layout}}$ for each image using GroundingDINO (Liu et al., 2024), based on the alignment between generated content and the provided bounding boxes. Let the resulting scores be sorted in descending order, and denote by $x_{(i)}$ the image with the $i$-th highest score.

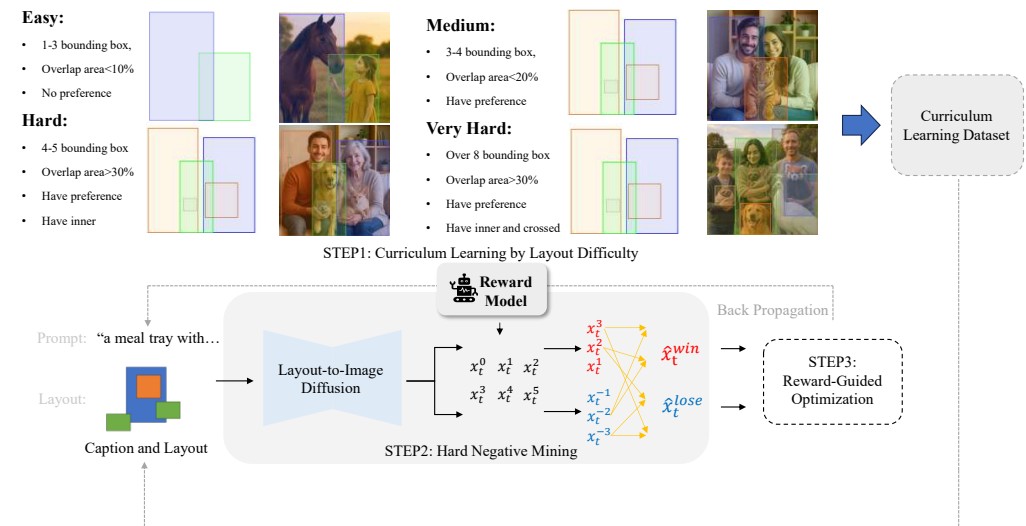

Figure 2: Overall pipeline of our curriculum learning and reward-guided optimization framework. Step 1 groups training samples into four difficulty levels (Easy, Medium, Hard, Very Hard) based on bounding-box number, overlap ratio, and preference conditions. Step 2 performs hard negative mining with a reward model to distinguish high- and low-quality generations. Step 3 applies reward-guided optimization to update the layout-to-image diffusion model, completing a closed training loop.

Based on this ranking, we construct a set of four distinct preference pairs:

$$\mathcal{P} = \big\{(x_{(1)}, x_{(6)}),\ (x_{(2)}, x_{(6)}),\ (x_{(1)}, x_{(5)}),\ (x_{(2)}, x_{(5)})\big\}, \quad |\mathcal{P}| = 4, \tag{3}$$

where each pair $(x^w, x^\ell)$ consists of a higher-scoring image $x^w$ (the *winner*) and a lower-scoring image $x^\ell$ (the *loser*). These pairs serve as supervised preference signals for fine-tuning.

This extreme pairing strategy provides two key advantages.

First, by forming several preference pairs within each mini-batch, our strategy uses the same inference budget more efficiently; the enlarged candidate pool yields a denser set of preference signals, providing richer and more informative supervision.

Second, extreme pairing amplifies both the layout reward gap and the log-probability difference under the current sampling policy:

$$\Delta \log p = \log p_\theta(x^w) - \log p_\theta(x^\ell),$$

which concentrates gradient updates on image pairs exhibiting the most significant spatial misalignment. Compared to random or arbitrary pairings, this targeted supervision produces stronger preference signals and accelerates convergence during fine-tuning.

### 3.2.3 STOCHASTIC SD-3 SAMPLER WITH EXPLICIT LOG-PROB

SD-3 (Esser et al., 2024) uses a deterministic Flow–ODE scheduler (Lipman et al., 2022) that converges to a unique solution for any fixed noise vector. To introduce stochasticity *while approximately preserving the sampling marginals* and to obtain *differentiable* action probabilities for RL, we extend the ODE

$$\mathrm{d}x_t = v_\theta(x_t, t)\,\mathrm{d}t$$

to an SDE under a *variance-time* parameterization:

$$\mathrm{d}x_t = v_\theta(x_t, t)\,\mathrm{d}t + \sigma(t)\,\mathrm{d}W_t,$$

where $x_t \in \mathbb{R}^D$, $v_\theta$ is the network-predicted velocity field, $\sigma(t) = \sqrt{1/\operatorname{SNR}(t)}$ follows the SD-3 log-SNR schedule, and $W_t$ is a standard Wiener process.

Define the cumulative variance

$$\Sigma(t) = \int_0^t \sigma^2(s)\,\mathrm{d}s,$$

and let $0 = t_0 < \cdots < t_N = 1$ be a partition of the time horizon with $\Sigma_i = \Sigma(t_i)$, $\Delta\Sigma_i = \Sigma_{i+1} - \Sigma_i$, and $\sigma_i = \sigma(t_i)$. A single Euler–Maruyama step (Lamba et al., 2007) over $[\Sigma_i, \Sigma_{i+1}]$ yields

$$x_{i+1} = x_i + \frac{\Delta\Sigma_i}{\sigma_i^2} v_\theta(x_i, t_i) + \sqrt{\Delta\Sigma_i}\,\varepsilon, \qquad \varepsilon \sim \mathcal{N}(0, I). \tag{4}$$

Because $\Sigma(t)$ is strictly increasing, reparameterizing time in $\Sigma$ changes step sizes but preserves the *Gaussian form* of the one-step transition, enabling controlled noise injection without redesigning the sampler.

To strengthen stochasticity, we scale the noise magnitude to $\hat{\sigma}_i = (1+\gamma)\sigma_i$ ($\gamma \geq 0$), which implies

$$\hat{\Delta\Sigma}_i = (1+\gamma)^2\,\Delta\Sigma_i. \tag{5}$$

In $\Sigma$-time this can be viewed as adjusting the step size while keeping the transition form unchanged. For small $\gamma$, the induced marginal shift is minor in practice (see Appendix and ablations); as $\gamma \to 0$, the sampler reduces to the original deterministic Flow–ODE.

The one-step transition admits a closed-form log-probability:

$$\log \pi_\theta(a_i = x_{i+1} \mid s_i = x_i) = -\frac{\|x_{i+1} - \mu_i\|^2}{2\,\Delta\Sigma_i} - \tfrac{1}{2}\log\big(2\pi\,\Delta\Sigma_i\big), \qquad \mu_i := x_i + \frac{\Delta\Sigma_i}{\sigma_i^2} v_\theta(x_i, t_i). \tag{6}$$

Differentiability flows through $\mu_i$ (and thus $v_\theta$), providing well-behaved action-probability estimates for preference-based and policy-gradient updates. When noise is disabled ($\Delta\Sigma_i = 0$), the variance term vanishes and the sampler collapses to the deterministic Flow–ODE. For numerical stability we use $\Delta\Sigma_i + \varepsilon$ (e.g., $10^{-6}$) in denominators and logarithms.

### 3.2.4 LOSS FUNCTION

To explicitly improve layout alignment without sacrificing fidelity or semantic consistency, we minimize the following preference-guided objective over preference pairs $(x_w, x_\ell)$ (constructed by the extreme pairing in Sec. 3.2.2):

$$\mathcal{L}_\theta = \mathbb{E}_{(x_w, x_\ell)\sim\mathcal{P}}\Big[ -\log\sigma\big(\beta\,\Delta\log p\big) + \lambda\big(1 - \sigma(\kappa\,\Delta r)\big)\tfrac{1}{2}\sum_{x\in\{x_w, x_\ell\}}\|\delta_\varepsilon(x)\|_2^2\Big]. \tag{7}$$

The first term is a pairwise logistic advantage that pushes the log-probability gap $\Delta\log p$ to increase in favor of the better-aligned sample (thus shifting probability mass toward high-IoU regions). The second term is an IoU-aware KL-style regularizer: when the preference gap $\Delta r$ is weak (i.e., $\sigma(\kappa\,\Delta r)$ is small), it pulls the current model toward a frozen reference to stabilize training. All symbols are defined *in situ* as

$$\Delta\log p = \log p_\theta(x_w) - \log p_\theta(x_\ell), \quad \Delta r = r_w - r_\ell, \quad \delta_\varepsilon(x) = \log p_\theta(x) - \log p_{\mathrm{ref}}(x), \tag{8}$$

where $r_i = R_{\mathrm{layout}}(x_i, \mathcal{M})$ is the layout reward, $\sigma(z) = 1/(1 + e^{-z})$ is the sigmoid, $p_\theta$ denotes the sampler-induced distribution (see Sec. 3.2.3), and $p_{\mathrm{ref}}$ is a frozen reference model. The scalars $\beta > 0$, $\kappa > 0$, and $\lambda \geq 0$ control the sharpness of the logistic term, the temperature on the reward gap, and the strength of the regularization, respectively.

This objective can be derived from the constrained problem

$$\max_\theta \mathbb{E}_{x\sim p_\theta}\big[R_{\mathrm{layout}}(x, \mathcal{M})\big] \quad \text{s.t.} \quad \mathrm{KL}\big(p_\theta\|p_{\mathrm{ref}}\big) \leq \varepsilon. \tag{9}$$

We approximate the expected reward by a weighted pairwise advantage $\mathbb{E}[\sigma(\kappa\,\Delta r)\,\Delta\log p]$, and locally approximate the KL by a second-order expansion $\mathrm{KL}(p_\theta\|p_{\mathrm{ref}}) \approx \tfrac{1}{2}\mathbb{E}_x\big[\|\delta_\varepsilon(x)\|_2^2\big]$ (up to constants), yielding Eq. equation 7.

This loss has two useful properties. First, the derivative of the logistic term with respect to the log-probability gap is strictly negative:

$$\frac{\partial \mathcal{L}_\theta}{\partial\,\Delta\log p} = -\beta\,\sigma\big(-\beta\,\Delta\log p\big) < 0, \tag{10}$$

so gradient-based updates monotonically increase $\Delta\log p$, raising the relative likelihood of high-IoU samples. Second, when the preference gap is weak ($\Delta r \to 0$), we have $\sigma(\kappa\,\Delta r) \to \tfrac{1}{2}$ and the

Table 1: Curriculum schedule of difficulty ratios for CreatiLayout and HicoNet across training iterations.

| Method | Iterations | Easy | Medium | Hard | Very Hard |
|---|---|---|---|---|---|
| CreatiLayout | 1–150 | 70% | 30% | 0% | 0% |
| | 150–400 | 30% | 60% | 10% | 0% |
| | 400–700 | 20% | 45% | 30% | 5% |
| | 700–1000 | 10% | 20% | 40% | 30% |
| HicoNet | 1–150 | 50% | 50% | 0% | 0% |
| | 150–400 | 20% | 50% | 30% | 0% |
| | 400–700 | 10% | 40% | 40% | 10% |
| | 700–1000 | 0% | 20% | 40% | 40% |

regularizer reduces to $\frac{\lambda}{2}\mathbb{E}_x[\|\delta_\varepsilon(x)\|_2^2]$, which suppresses semantic drift and preserves image quality under noisy or uninformative comparisons.

Overall, combining the pairwise logistic advantage with the IoU-aware regularizer provides a learning signal that is both directional and stable; in our experiments (Sec. 4.1.2), this objective improves expected IoU without degrading fidelity or semantic consistency.

## 4 EXPERIMENTS

### 4.1 EXPERIMENT SETUP

#### 4.1.1 BASELINES

To explore the effectiveness and generalizability of our approach across different diffusion backbones, we implement and fine-tune two model variants. Specifically, **HiCoNet-Layout** (Cheng et al., 2024) is built upon Stable Diffusion 1.5, a widely adopted open-source architecture known for its balance between generation quality and computational efficiency. In contrast, **CreatiLayout** (Zhang et al., 2024) leverages Stable Diffusion 3, a more recent and powerful backbone offering enhanced semantic understanding and generation fidelity. This dual-backbone setup allows us to evaluate how our layout generation method adapts to varying model capacities and architectural changes, and to verify whether the improvements introduced by our approach consistently hold across diffusion model versions.

#### 4.1.2 DATASET SCHEDULING

We adopt a *stage-wise mixed curriculum* covering four difficulty levels (Easy / Medium / Hard / Very Hard). We first construct four pools from *real* layouts (7,000 samples each). At the beginning of each training stage, we resample with replacement according to the preset ratios, shuffle, and form a 10,000-sample subset for that stage; within the stage, this subset is reshuffled every epoch. At the next stage, a new 10,000-sample subset is re-sampled under the updated ratios. The concrete categorization and schedule are provided in Table 1.

#### 4.1.3 TRAINING DETAILS

**Data construction.** Unlike traditional supervised fine-tuning, **RLLay** does not rely on a pre-fixed corpus of real image–layout pairs. To keep the training distribution close to real scenes, we *do not synthesize* bounding boxes; instead, we *select real layouts* directly from the annotations of LAYOUTSAM and COCO 2017 (Lin et al., 2014). According to our difficulty criteria (object count, overlap, and preferences/relations), we construct four difficulty pools—*Easy / Medium / Hard / Very Hard*—each containing **7,000** layout samples (preserving the original categories and text descriptions). At each stage of the curriculum, we resample and mix from the four pools according to the stage-specific ratios to form a **10,000**-sample training subset for that stage.

**Evaluation protocol.** We evaluate on two dataset families: the fine-grained, open-set LAYOUTSAM and the coarse-grained, closed-set COCO 2017. **LayoutSAM-Eval:** following (Zhang et al., 2024), we evaluate on LAYOUTSAM-EVAL, which contains **5,000** prompts with metadata (category and

Table 2: Overall performance of RLLay compared to existing state-of-the-art approaches on the LayoutSAM-Eval(Zhang et al., 2024) benchmark.

| Method | Spatial | | | | Quality | | Accuracy |
|---|---|---|---|---|---|---|---|
| | IoU ↑ | AP↑ | AP50↑ | AR↑ | FID ↓ | CLIP-Score(Global)↑ | VQA↑ |
| GLIGEN | 60.09 | 18.34 | 37.18 | 29.77 | 25.93 | - | - |
| Instance Diffusion | 77.03 | 37.16 | 46.55 | 40.10 | 23.16 | 27.70 | - |
| HiCO | 72.05 | 29.42 | 41.55 | 33.50 | 18.64 | 29.52 | 91.45 |
| HiCO + RLLay | 80.12 | 38.40 | 47.75 | 41.30 | 17.12 | 29.03 | 93.18 |
| CreatiLayout | 68.01 | 29.55 | 50.17 | 33.20 | 15.26 | 27.89 | 82.26 |
| CreatiLayout + RLLay | 74.97 | 33.15 | 55.37 | 35.60 | 13.27 | 27.31 | 86.08 |

layout annotations). **COCO-Eval:** following prior work (Yang et al., 2023a; Zheng et al., 2023), we filter out images with too small effective area and too few bounding boxes from the COCO validation set, resulting in **4,652** validation images.

**Metrics.** To assess *layout compliance*, we use GROUNDINGDINO (Liu et al., 2024) to detect objects in generated images and match them with the ground-truth boxes, computing standard **IoU**, **AP**, **AP50**, and **AR**. To measure *overall generation quality*, we report **FID** (Heusel et al., 2017) and **CLIP-Score** (Radford et al., 2021). We also report a **VQA** score (based on a visual question answering model that answers prompt-related questions) to assess the *semantic accuracy* of the generations.

**Training setup.** We use the ADAMW optimizer (Loshchilov & Hutter, 2017) with distributed training on **8× NVIDIA A6000** GPUs; the local batch size is **32 preference pairs per GPU**, and training runs for **1000 Iterations**. Resolutions are model-specific: HICONET is trained at $512 \times 512$, whereas CREATILAYOUT is trained at $1024 \times 1024$.

## 4.2 RESULTS

### 4.2.1 EVALUATION ON LAYOUTSAM-EVAL

Table 2 reports quantitative results on the LayoutSAM-Eval (Zhang et al., 2024) benchmark. Our reinforcement learning framework RLLay, consistently improves both spatial and visual quality metrics across two strong baselines. When applied to HiCO, RLLay yields clear gains on all spatial measures—IoU, AP, AP50, and AR—while further lowering FID and enhancing semantic accuracy (VQA). A similar trend is observed on CreatiLayout, where RLLay substantially boosts localization accuracy (IoU and AP50) and reduces FID, confirming that the generated layouts and final images are both more precise and more visually faithful. Compared with diffusion-based methods such as GLIGEN and Instance Diffusion, RLLay-enhanced models not only maintain competitive generative quality but also deliver markedly better alignment between textual descriptions and visual layouts, as reflected in higher CLIP-Score and VQA. These results demonstrate that the proposed reinforcement-guided optimization effectively strengthens spatial reasoning while preserving high-fidelity image synthesis.

### 4.2.2 EVALUATION ON COCO

Table 3 presents a comprehensive comparison of RLLay with representative state-of-the-art layout-to-image generation approaches on the COCO (Lin et al., 2014) benchmark. Our method consistently achieves superior spatial alignment, as reflected by higher IoU, AP, AP50, and AR scores, while maintaining strong perceptual quality (lower FID and higher CLIP-Score) and semantic accuracy (higher VQA). Compared with diffusion-based baselines such as GLIGEN and Instance Diffusion, RLLay delivers more precise object localization and competitive visual fidelity. Relative to reinforcement-learning-enhanced methods (e.g., HiCO + RLLay), our approach further improves detection metrics and semantic faithfulness. In particular, integrating our reward-guided curriculum with CreatiLayout (CreatiLayout + RLLay) yields notable gains across all three categories of metrics, confirming the effectiveness of our design in balancing spatial control and generative quality. Noted that CreatiLayout

Table 3: Overall performance of RLLay compared to existing state-of-the-art approaches on the COCO(Lin et al., 2014) benchmark.

| Method | Spatial | | | | Quality | | Accuracy |
|---|---|---|---|---|---|---|---|
| | IoU ↑ | AP↑ | AP50↑ | AR↑ | FID ↓ | CLIP-Score(Global)↑ | VQA↑ |
| GLIGEN | 51.60 | 19.48 | 32.10 | 30.68 | 27.94 | 21.63 | 68.47 |
| Instance Diffusion | 84.40 | 38.91 | 47.60 | 43.10 | 25.66 | 25.65 | 79.49 |
| HiCO | 70.57 | 22.12 | 33.26 | 29.00 | 26.05 | 27.17 | 75.12 |
| HiCO + RLLay | 78.29 | 37.33 | 48.19 | 44.30 | 25.03 | 26.98 | 81.98 |
| CreatiLayout | 56.59 | 12.09 | 26.08 | 17.60 | 26.18 | 26.19 | 69.74 |
| CreatiLayout + RLLay | 62.58 | 15.86 | 32.19 | 18.30 | 21.09 | 25.14 | 75.54 |

Table 4: Ablation study on the COCO Benchmark for **CreatiLayout** and **HicoNet**. Metrics include AP, AP50, AR, FID, CLIP-Score(Global), and IoU.

| Method | AP ↑ | AP50 ↑ | AR ↑ | FID ↓ | CLIP-Score(Global) ↑ | IoU ↑ |
|---|---|---|---|---|---|---|
| CreatiLayout | 12.09 | 26.08 | 17.60 | 26.18 | 26.19 | 56.69 |
| + DPO | 12.87 | 27.35 | 18.20 | 26.01 | 25.79 | 58.35 |
| + DSPO | 15.37 | 30.93 | 18.20 | 21.96 | 25.76 | 60.46 |
| + GRPO | 15.72 | 31.72 | 18.30 | 26.25 | 22.63 | 61.33 |
| + ARPO | 15.86 | 32.19 | 18.30 | 21.09 | 25.14 | 62.58 |
| HicoNet | 22.12 | 33.26 | 29.00 | 26.05 | 27.17 | 70.57 |
| + DSPO | 29.59 | 41.24 | 35.80 | 25.23 | 26.85 | 75.85 |
| + GRPO | 29.45 | 40.60 | 35.70 | 29.35 | 25.23 | 74.95 |
| + ARPO | 37.33 | 48.19 | 44.30 | 25.03 | 26.80 | 78.29 |

does not release the version after finetuning on the COCO dataset, leading to lower performance than the numbers reported in its original paper.

### 4.2.3 ABLATION STUDY

Table 4 reports the quantitative comparison on the COCO benchmark for CreatiLayout and HicoNet. The proposed reinforcement learning method (ARPO) consistently improves both spatial and visual quality metrics over all baselines. For CreatiLayout, ARPO achieves the best spatial alignment, raising AP and AP50 to 15.86 and 32.19 while increasing IoU to 62.58, and simultaneously lowering FID to 21.09, indicating sharper and more faithful image synthesis. A similar trend is observed for HicoNet, where ARPO delivers substantial gains across detection-oriented metrics (AP from 22.12 to 37.33, AP50 from 33.26 to 48.19, AR from 29.00 to 44.30) while also reducing FID and maintaining competitive CLIP-Score. Compared with other reinforcement-preference optimization approaches such as DPO, DSPO, and GRPO, our method provides a more balanced improvement on both spatial precision and visual fidelity, confirming its effectiveness in aligning generated layouts and images with the intended semantics

## 5 CONCLUSION

In this work, we presented **RLLay**, a reinforcement-learning-driven pipeline for preference-based optimization of layout-to-image diffusion models. Unlike conventional methods that rely heavily on paired layout–image datasets, **RLLay** leverages pairwise human or model feedback to guide policy updates, significantly reducing data collection requirements while improving generative alignment. Built upon our novel reinforcement learning algorithm **ARPO**, the framework integrates curriculum scheduling, hard negative mining, and preference-guided reward modeling to achieve stable and sample-efficient optimization. Extensive experiments on COCO and LayoutSAM-Eval benchmarks demonstrate consistent improvements in both spatial accuracy and perceptual quality over strong diffusion and preference-optimization baselines, including DPO, DSPO, and GRPO. These results highlight the effectiveness and generality of our approach, opening avenues for broader applications of reinforcement preference optimization in complex generative modeling tasks.

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

# A APPENDIX

## A.1 ETHICS STATEMENT

This work focuses on reinforcement preference optimization for layout-to-image diffusion models and evaluates the proposed RLLay pipeline on publicly available benchmarks (e.g., COCO and LayoutSAM-Eval). All datasets are released for research use and comply with their respective licenses and privacy regulations. We are aware that generative vision–language systems can amplify societal biases or produce misleading imagery. To mitigate these concerns, we adopted established filtering procedures, reported results across diverse metrics, and carefully inspected generated samples. We stress that our model is intended solely for academic and constructive applications and should not be used to create harmful, discriminatory, or deceptive content.

## A.2 REPRODUCIBILITY STATEMENT

We have taken comprehensive steps to ensure reproducibility. Full implementation details—including network architecture, reward model design, the proposed ARPO algorithm, training schedules (curriculum stages, hyperparameters), and hardware configurations—are provided in the main text and the supplementary material. All evaluation protocols and benchmark settings are described in the experiment section. We will release the source code, pretrained weights, and configuration scripts upon publication to facilitate independent verification and future research.

## A.3 THE USE OF LARGE LANGUAGE MODELS

Large language models (LLMs) were employed only for limited editorial assistance, such as grammar checking and minor language polishing. They were not used to generate scientific ideas, design experiments, analyze results, or write substantive technical content. The authors assume full responsibility for every aspect of the work, and LLMs are not considered contributors or authors.

## A.4 ADDITIONAL NOTES ON THE RANDOM SD-3 SAMPLER

**What "approximately preserving marginals" means.** Extending the Flow–ODE to a variance-time SDE does not *strictly* guarantee identical marginals to the original ODE. "Approximately preserving" indicates that: (i) as the noise scaling $\gamma \to 0$ and the step size $\Delta\Sigma_i \to 0$, the local Markov kernel of the SDE matches the probability flow of the ODE to first order, so the shift in the generated distribution is controlled; (ii) with finite steps and a small $\gamma$, empirical changes in image quality (e.g., FID/CLIP) and alignment (e.g., IoU/AP) remain acceptable. Increasing $\gamma$ raises stochasticity and exploration but also enlarges marginal drift and quality variance.

**Why variance-time reparameterization.** In $\Sigma$-time the one-step transition is explicitly Gaussian:

$$x_{i+1} \sim \mathcal{N}\big(\mu_i,\ \Delta\Sigma_i\, I\big), \qquad \mu_i \;=\; x_i + \frac{\Delta\Sigma_i}{\sigma_i^2}\, v_\theta(x_i, t_i),$$

which yields two practical benefits: (i) a closed-form one-step log-probability $\log \pi_\theta(x_{i+1}\,|\,x_i)$ that backpropagates through $v_\theta$; (ii) fine control of the exploration strength by tuning $\Delta\Sigma_i$ and $\gamma$ without changing the Gaussian form.

**Accumulating trajectory log-probabilities.** While the main text gives the one-step log-probability, preference/policy learning uses the *trajectory* log-probability via stepwise accumulation:

$$\log p_\theta(x_{0:N}) \;=\; \sum_{i=0}^{N-1} \log \pi_\theta(x_{i+1}\,|\,x_i) \;+\; \log p(x_0),$$

where $\log p(x_0)$ is an initialization prior (constant or negligible). The preference gap $\Delta \log p$ and the reference difference $\log p_\theta - \log p_{\mathrm{ref}}$ are accumulated in the same way, which is numerically more robust than a single-step proxy.

**Choosing and scheduling $\gamma$.** *Fixed small values:* $\gamma \in [0.05, 0.20]$ often strikes a good balance between having differentiable action probabilities and limiting distributional drift. *Piecewise/annealed:* use larger $\gamma$ early for exploration/separability and reduce it later to recover quality. *Step-adaptive:* employ smaller $\gamma$ at larger $\Delta\Sigma_i$ to avoid excessive variance accumulation.

**Numerical stability and implementation tips.** Use $(\Delta\Sigma_i + \varepsilon)$ (e.g., $\varepsilon = 10^{-6}$) in denominators and logarithms to suppress blow-ups; compute $\|x_{i+1} - \mu_i\|^2$ in at least `float32` and optionally normalize by feature dimension. Ensure that sampling and $\log \pi_\theta$ share the same random seed and schedule to avoid "mismatch" bias. For long trajectories, gradient clipping and an EMA teacher help stabilize training.

**Relationship to probability-flow ODE.** The probability-flow ODE (PF-ODE) gives deterministic trajectories in the zero-diffusion limit. With small $\gamma$ and small steps, the SDE's drift (mean update) aligns with PF-ODE locally, while the added diffusion supplies a stochastic policy required by RL. "Preserving marginals" here should be understood as a *local approximation* and *empirically controlled* behavior, rather than an exact equivalence.

**Compatibility with guidance and conditioning.** *Classifier-free guidance (CFG)* remains applicable; $v_\theta$ in $\mu_i$ simply includes the guided drift. *Text/layout conditioning* only enters through $v_\theta(x_i, t_i; \text{cond})$, to which the SDE design is agnostic. *Reference model* log-probabilities should reuse the same discretization and noise draws to reduce variance and enable effective differencing.

**Discretization and number of steps.** A larger number of steps $N$ approaches the continuous limit but increases compute linearly; in practice $N$ is set equal to (or slightly larger than) the original SD-3 inference steps. Non-uniform $\Sigma$-grids are recommended: densify segments where the log-SNR schedule is most sensitive, which improves the stability of $\log \pi_\theta$ estimation.

