# OpenReview forum: "Reinforce Your Layout: Online Reward-Guided Diffusion for Layout-to-lmage Generation"
_ICLR.cc/2026/Conference — ICLR 2026 Conference Withdrawn Submission_

### Official Review · Reviewer_fnUK · 2025-10-29

**Soundness:** 1
**Presentation:** 2
**Contribution:** 2
**Rating:** 2
**Confidence:** 5

**Summary:**

This work studies the layout-to-image generation problem. The main efforts in this work is to use the reinforcement learning to improve the layout-to-image generation performance. The proposed method applies the online policy training strategy on a pre-trained layout-to-image generation model. The author first rank the difficulties of the samples into four levels and uses a curriculum training scheme by starting from easy samples to very hard samples. In addition, the author proposes to use SDE instead of ODE during the sampling of RL candidates. Afterward, the model is optimized by a custom RL loss. In the experiment, the author shows that the proposed approach can be applied on different layout-to-image model and the results show that the performance on LayoutSAM-Eval benchmark has improved.

**Strengths:**

* This work studies the application of reinforcement learning specific for layout-to-image generation, which can be a promising direction for layout-to-image generation quality improvement.
* The proposed method has been applied on two different layout-to-image models, indicating it generalizability.

**Weaknesses:**

Major issues:
* The reviewer is quite confused about the motivation part in sec 3.2.1. For example:
    * “base model fails to satisfy complex layout constrains, so parallel sampling yields candidate pairs of uniformly poor quality with barely separable preference/reward signals, making policy improvement difficult.” What kind of complex layout constrain does base model usually to fail and has the proposed method efficiently address these complex layout generation? And why does uniformly poor quality makes policy improvement difficult? Is there any evidence that current L2I provides uniformly poor quality and the policy improvement is difficult?
    * “the wide distributional span within a single dataset further amplifies training instability.” The reviewer does not understand the relation between the data distribution span and the training instability. And why does the curriculum training is a suitable solution for this problem.
* The methodology of the proposed method is not well elaborated. For example
    * In the difficulty level definitions, what does “no preferences/relations” mean? How is containment and overlap defined (like instance wise or region size wise).
    * The author claims in ln 206-209 that “policy continues to improve under heavy overlap and complex relations without collapsing.” The reviewer wonders is there any evidence that the audience can see why non-curriculum training can collapse.
    * The current rewards is purely based on the IoU, but in the L2I problem, the object fidelity is also important. The reviewer wonders why the author designed reward solely based on IoU.
    * The sample pairing in Eq 3 is also proposed without a sufficient justification.
* The reviewer also has questions on the sampling modification in Sec 3.2.3. It seems the SDE adopted in ln 264 is just the ODE + adding gaussian noise. This would be a naive fall back from simplified ODE back to the original DDPM’s SDE. However, there are quite a few heuristic designs added here, like Eq5 and Eq6. The reviewer cannot quite be convinced by the propose solutions. In addition, the ln 275-277 is confusing, the reviewer does not understand what this sentence is conveying. Moreover, the ln 289-290 that Eq6 is “providing well-behaved action-probability estimates for preference-based and policy-gradient updates.”, but the reviewer cannot find any evidence to support this statement.
* On the loss function in Sec 3.2.4, the analysis in Eq 10 seems to ignore the second term in the Eq 7. There are two issues here:
    * Even if ignoring the second term in Eq 7, the derived partial derivative show be -\beta[1 - \sigma(-\beta \Delta log p)].
    * The reviewer does not think the second term in Eq 7 should be ignored. It seems here the author’s ignoring of second term is based on the use of the approximation in ln 316. However, if use the original expansion of KL, there will be one more \Delta log p term in the second part. The reviewer is not convinced on using the approximation and treat it as a constant independent of  \Delta log p.
* There are plenty of things proposed in this work but the corresponding discussion are quite scarce. Currently there are so many factors that it makes the reviewer very confused where the most important performance gain is from. However, the current ablation study is only comparing against to other RL strategy and how these RL strategies are applied in the L2I problem is also not clear in the manuscript.
* The chose baselines is just Instance Diffusion and it has been released quite long ago, the reviewer suggests the author adding more comparison to more recent works on L2I problem. Also, the VQA accuracy for GLIGEN and Instance Diffusion is missing in Table 2.

Minor issues:
* As a CV paper, the reviewer would have expectation to see visual comparison against existing methods to show the advantages of the proposed method, and the effectiveness of each proposed componets, but the current manuscript provides insufficient visual results.
* Some of the equations are missing numbers.

**Questions:**

Please see my weaknesses

---

> ### Author Response · Authors · 2025-11-13
>
> Dear Reviewer,
>
> Thank you very much for your careful reading of our submission and for providing such detailed and objective comments. We genuinely appreciate the time and effort you invested in pointing out unclear motivations, methodological gaps, and missing analyses.
>
> Given the extent of the revisions required, we have decided to withdraw this version of the paper and substantially improve the work before resubmitting it to another venue. In particular, we will (i) expand the set of baselines to include more recent layout-to-image methods, (ii) conduct more fine-grained ablation studies to disentangle the contributions of curriculum design, pairing strategy, and sampling/log-prob estimation, and (iii) add richer qualitative visualizations and more thorough discussions to clearly illustrate where the performance gains come from. We will also carefully revise the sections whose motivation or theoretical justification was unclear (e.g., curriculum scheduling, SDE-based sampling, and loss derivations), and provide additional empirical evidence wherever we made strong claims.
>
> Your feedback has been extremely helpful in identifying the weak points of our current draft and will directly guide the next iteration of this project. Thank you again for your constructive and thoughtful review.
>
> Best regards,
> The authors

---

### Official Review · Reviewer_XTmo · 2025-10-30

**Soundness:** 2
**Presentation:** 2
**Contribution:** 3
**Rating:** 4
**Confidence:** 4

**Summary:**

The paper presents RLLay, an online RL framework for layout-to-image generation that fine-tunes diffusion models to better follow spatial layouts. Built on a new reinforcement learning algorithm ARPO, it uses curriculum scheduling, hard-negative mining, and preference-guided rewards for stable, sample-efficient training.

**Strengths:**

1. To the best of my knowledge, this is the first RL-based pipeline for layout-toimage generation.

2. Extending the flow ODE to a variance-time SDE (with stability notes) to enable preference/RL learning is a nice technical touch.

**Weaknesses:**

1. Table 1 is somewhat confusing, as the process for determining these ratios is not described. The settings for CreatiLayout and HicoNet differ, suggesting that the ratios were not selected arbitrarily, yet the paper offers no explanation for the rationale behind these choices.

2. The paper employs a “from easy to hard” curriculum strategy based on layout complexity, aiming to progressively train the model to handle increasingly complex layouts. While this approach appears reasonable, the experiments do not include per-difficulty comparisons against baselines to validate whether the method indeed improves performance on harder layout constraints.

3. The work lacks several important ablations: (1) an analysis of how varying the number of extreme preference pairs per (prompt, layout) affects results, and (2) curriculum learning ablations—such as experiments without curriculum learning or with different difficulty boundaries.

4. Compared with the selected baseline, Instance Diffusion, the reported improvements are limited; moreover, the Intersection-over-Union (IoU) score on the COCO benchmark is notably lower.

**Questions:**

1. Figure 1 contains several noticeable issues. In the first row, the layout appears to omit a bounding box for the notebook; the green box should be labeled “desk lamp”; and the capitalization of “Pen holder” is inconsistent with the other category labels. Furthermore, including the corresponding real image (i.e., the source of this layout) side by side would make the visualization clearer and the comparison more intuitive.

2. It is unclear whether the proposed approach supports multi-instance layouts—for example, a layout containing multiple instances of the same object, such as five cats. The paper should clarify whether the method can accurately interpret and generate such cases, and ideally include experiments or visual examples demonstrating its capability in handling multi-instance scenarios.

---

> ### Author Response · Authors · 2025-11-13
>
> Dear Reviewer,
>
> Thank you very much for your careful reading of our submission and for providing such detailed and objective comments. We genuinely appreciate the time and effort you invested in pointing out unclear motivations, methodological gaps, and missing analyses.
>
> Given the extent of the revisions required, we have decided to withdraw this version of the paper and substantially improve the work before resubmitting it to another venue. In particular, we will (i) expand the set of baselines to include more recent layout-to-image methods, (ii) conduct more fine-grained ablation studies to disentangle the contributions of curriculum design, pairing strategy, and sampling/log-prob estimation, and (iii) add richer qualitative visualizations and more thorough discussions to clearly illustrate where the performance gains come from. We will also carefully revise the sections whose motivation or theoretical justification was unclear (e.g., curriculum scheduling, SDE-based sampling, and loss derivations), and provide additional empirical evidence wherever we made strong claims.
>
> Your feedback has been extremely helpful in identifying the weak points of our current draft and will directly guide the next iteration of this project. Thank you again for your constructive and thoughtful review.
>
> Best regards,
> The authors

---

### Official Review · Reviewer_TmJD · 2025-10-31

**Soundness:** 3
**Presentation:** 2
**Contribution:** 2
**Rating:** 4
**Confidence:** 4

**Summary:**

This paper proposes RLLay, an online reinforcement learning framework for layout-to-image generation, aiming to improve spatial alignment between generated images and target layouts. The key technical component is ARPO (Advanced Relative Policy Optimization)—a pairwise preference-based RL algorithm that fine-tunes diffusion models through online sampling and ranking by a layout reward (mean IoU).
RLLay further incorporates (1) curriculum learning by layout difficulty to stabilize online optimization, and (2) hard negative mining to construct more informative preference pairs.
Experiments on LayoutSAM-Eval and COCO benchmarks show consistent improvement in spatial accuracy and fidelity over diffusion and preference-optimization baselines such as DPO, DSPO, and GRPO.

**Strengths:**

1. The staged “easy-to-hard” layout scheduling is practical and empirically improves stability—a rare but useful consideration in RL-based diffusion.
2. Table 4 clearly compares DPO, DSPO, GRPO, and ARPO, providing quantitative evidence of the proposed algorithm’s effectiveness.

**Weaknesses:**

1. The “ARPO” update essentially combines a logistic pairwise loss from DPO-style optimization with a KL regularizer, which makes it conceptually close to DSPO and GRPO. The paper claims that ARPO brings stability advantages, but it does not provide either theoretical justification or empirical evidence (e.g., variance plots or training dynamics) to support this claim.

2. The related work section misses several latest layout-to-image generation studies, such as 3DIS-FLUX, as well as other recent approaches that incorporate curriculum learning, which has become a common and effective strategy in multimodal understanding. A more comprehensive discussion of these works would better contextualize the contribution.

3. In Figure 2, the examples of “Medium,” “Hard,” and “Very Hard” layouts appear almost identical, and the “Very Hard” case shows a layout that does not match the target image. It is unclear whether these examples are actual training samples or illustrative placeholders. Please clarify the data source or design intention here.

4. The level definitions described in the main text are inconsistent with those shown in Figure 2. This inconsistency should be resolved to ensure clarity about how difficulty levels are defined and applied in practice.

5. The visualizations and quantitative benchmarks mostly involve layouts with five or fewer objects (see Fig. 1 and Sec. 4.1.2). Since the paper’s main claim concerns improved layout fidelity, it would be valuable to include evaluations on denser or more occluded scenes (e.g., ≥ 10 objects or high-overlap layouts) to better demonstrate robustness.

6. The reward function relies solely on IoU, which measures spatial overlap but overlooks other important aspects such as appearance consistency (e.g., object size, color, texture, and contextual realism). Incorporating more comprehensive reward components could lead to more balanced improvements.

7. All reported evaluations depend entirely on automatic metrics (IoU, FID, CLIP, VQA). Given that the paper focuses on preference alignment, it would be helpful to include a small-scale user study or human evaluation comparing perceptual layout quality to strengthen the empirical claims.

8. The paper lacks qualitative analyses—there are no visual comparisons or case studies illustrating how RLLay improves generation quality beyond numerical metrics.

**Questions:**

see weakness.

---

> ### Author Response · Authors · 2025-11-13
>
> Dear Reviewer,
>
> Thank you very much for your careful reading of our submission and for providing such detailed and objective comments. We genuinely appreciate the time and effort you invested in pointing out unclear motivations, methodological gaps, and missing analyses.
>
> Given the extent of the revisions required, we have decided to withdraw this version of the paper and substantially improve the work before resubmitting it to another venue. In particular, we will (i) expand the set of baselines to include more recent layout-to-image methods, (ii) conduct more fine-grained ablation studies to disentangle the contributions of curriculum design, pairing strategy, and sampling/log-prob estimation, and (iii) add richer qualitative visualizations and more thorough discussions to clearly illustrate where the performance gains come from. We will also carefully revise the sections whose motivation or theoretical justification was unclear (e.g., curriculum scheduling, SDE-based sampling, and loss derivations), and provide additional empirical evidence wherever we made strong claims.
>
> Your feedback has been extremely helpful in identifying the weak points of our current draft and will directly guide the next iteration of this project. Thank you again for your constructive and thoughtful review.
>
> Best regards,
> The authors

---

### Official Review · Reviewer_8bi3 · 2025-11-01

**Soundness:** 3
**Presentation:** 3
**Contribution:** 3
**Rating:** 6
**Confidence:** 3

**Summary:**

This paper introduces RLLay, an online reinforcement-learning framework for layout-to-image diffusion models. It directly optimizes layout fidelity using an IoU-based reward with online policy updates, combining curriculum learning, hard negative mining, and a stochastic extension of SD-3 sampling for differentiable log-probabilities. Experiments on LAYOUTSAM-Eval and COCO show consistent improvements in spatial alignment, fidelity, and semantic accuracy over diffusion and RL-based baselines.

**Strengths:**

[+] RLLay directly optimizes layout fidelity with a clear IoU-based reward, improving spatial control without auxiliary guidance.

[+] The easy-to-hard curriculum and extreme pairing strategy stabilize training and enhance gradient efficiency.

[+] Strong baselines (HiCO, CreatiLayout, DPO, DSPO, GRPO) and metrics show consistent quantitative and qualitative gains.

[+] Figures and tables are well-organized and visually support the claims.

**Weaknesses:**

[-] Heavy reliance on GroundingDINO may introduce bias, but robustness to detection errors is not analyzed.

[-] Effects of λ, β, and κ on stability and alignment are not systematically studied.

[-] Missing analyses on out-of-distribution layouts or potential reward overfitting.

**Questions:**

1. How sensitive is RLLay to the accuracy of the GroundingDINO reward model?
1. Please clarify how trajectory-level log-probabilities are computed and accumulated.
1. Are there qualitative cases where optimizing IoU leads to perceptual artifacts or “metric gaming”?

---

> ### Author Response · Authors · 2025-11-13
>
> Dear Reviewer,
>
> Thank you very much for your careful reading of our submission and for providing such detailed and objective comments. We genuinely appreciate the time and effort you invested in pointing out unclear motivations, methodological gaps, and missing analyses.
>
> Given the extent of the revisions required, we have decided to withdraw this version of the paper and substantially improve the work before resubmitting it to another venue. In particular, we will (i) expand the set of baselines to include more recent layout-to-image methods, (ii) conduct more fine-grained ablation studies to disentangle the contributions of curriculum design, pairing strategy, and sampling/log-prob estimation, and (iii) add richer qualitative visualizations and more thorough discussions to clearly illustrate where the performance gains come from. We will also carefully revise the sections whose motivation or theoretical justification was unclear (e.g., curriculum scheduling, SDE-based sampling, and loss derivations), and provide additional empirical evidence wherever we made strong claims.
>
> Your feedback has been extremely helpful in identifying the weak points of our current draft and will directly guide the next iteration of this project. Thank you again for your constructive and thoughtful review.
>
> Best regards,
> The authors

---

### Note · Authors · 2025-11-13

I have read and agree with the venue's withdrawal policy on behalf of myself and my co-authors.